# Real-Time Deep Recognition of Standardized Liver Ultrasound Scan Locations

**DOI:** 10.3390/s23104850

**Published:** 2023-05-17

**Authors:** Jonghwan Shin, Sukhan Lee, Juneho Yi

**Affiliations:** 1Department of Electrical and Computer Engineering, Sungkyunkwan University, Suwon 16419, Republic of Korea; hwanyy@skku.edu (J.S.); jhyi@skku.edu (J.Y.); 2Artificial Intelligence Department, Sungkyunkwan University, Suwon 16419, Republic of Korea

**Keywords:** ultrasound image, liver scan, deep learning, hierarchical classification

## Abstract

Liver ultrasound (US) plays a critical role in diagnosing liver diseases. However, it is often difficult for examiners to accurately identify the liver segments captured in US images due to patient variability and the complexity of US images. Our study aim is automatic, real-time recognition of standardized US scans coordinated with reference liver segments to guide examiners. We propose a novel deep hierarchical architecture for classifying liver US images into 11 standardized US scans, which has yet to be properly established due to excessive variability and complexity. We address this problem based on a hierarchical classification of 11 US scans with different features applied to individual hierarchies as well as a novel feature space proximity analysis for handling ambiguous US images. Experiments were performed using US image datasets obtained from a hospital setting. To evaluate the performance under patient variability, we separated the training and testing datasets into distinct patient groups. The experimental results show that the proposed method achieved an F1-score of more than 93%, which is more than sufficient for a tool to guide examiners. The superior performance of the proposed hierarchical architecture was demonstrated by comparing its performance with that of non-hierarchical architecture.

## 1. Introduction

Liver disease is a major global health concern that affects millions of people worldwide. The early detection and treatment of liver disease is crucial for improving patient outcomes and preventing complications. Ultrasound (US) is a widely used imaging technique for liver disease diagnosis because it is non-invasive, safe, and cost-effective. Ultrasonography offers several advantages over other imaging techniques, such as computed tomography and magnetic resonance imaging, including the absence of radiation, affordability, and portability [1]. However, identifying key liver locations and monitoring the progress of US examinations can be challenging due to variability in the appearance of US images between individuals. Therefore, a system is needed to directly guide examiners to key liver regions during a US examination. Such a system could both be a valuable tool for medical education and help reduce variability in the diagnostic precision of skilled and unskilled examiners. Furthermore, AI-based computer-aided diagnosis tools can play an important role in assisting doctors, especially in remote areas with a lack of medical support [2].

However, this task is challenging due to the variability in US images and the complexity of liver vessel morphology, which plays a crucial role in accurately localizing liver sites. Although providing segmentation information about which organs are being examined can be helpful in diagnosis, it is challenging to precisely locate the region being examined in real space [3,4,5,6].

Recent studies have proposed using convolutional neural network (CNN)-based methods to automatically identify major liver cross-sections [7]. Our proposed method differs from the existing approaches that use pre-trained CNNs, such as the automatic classification of liver US standard plane (LUSP) images. The LUSP classification sorts images by focusing on specific image features. However, classifying the deformations that slight motions can cause in US images can be challenging. Research that focuses on US image registration could help with that problem [8], but again, because those studies focus on image features, their performance can be poor in the presence of large motion. Our proposed approach has a slightly different goal. We emphasize the relationship between the liver and the probe posture to better guide probe scanning during diagnosis and ameliorate the problem of subject variation by evaluating a new subject. Nonetheless, all CNN-based methods have limitations in capturing long-range dependencies between image features, which can limit their accuracy in some applications. Moreover, those methods do not always generalize well across different subject datasets due to variability in US images, which can limit their clinical utility [9].

In contrast, transformer models, which are based on self-attention mechanisms, have emerged as state-of-the-art models in various fields, such as natural language processing and computer vision [10]. In this study, we use a transformer decoder-based segmentation network (MaskFormer) [11] that has shown promising results in addressing the challenges of image segmentation, particularly in the context of liver US images [5].

Our main contribution is proposing a location-guided US system for accurate liver diagnosis that emphasizes the probe-posture and liver relationship. The use of a segmentation network also contributes to better generalizability by considering the differences between the training data and the individuals used in the testing data. By focusing on the relationship between the probe posture and the liver anatomy, we can better guide examiners during the diagnostic process, making it easier for them to navigate and identify key liver regions. Furthermore, our system incorporates vessel segmentation information to increase the accuracy of US examinations, especially in cases in which precise localization is essential.

Recent advances in deep learning approaches, such as few-shot learning and zero-shot learning [12], have been used in medical image classification to learn from sparse data and handle class imbalance [13]. Generative adversarial networks have also been proposed to generate realistic data that can supplement an original dataset, and other researchers have used domain adaptation techniques to handle dataset shifts [14,15]. Those studies highlight the importance of developing new techniques and approaches to handle challenging and ambiguous data in medical image classification tasks. In our work, we introduce a method for handling ambiguous data caused by slight movements during scanning, which can cause uncertainty in determining the correct diagnostic location class. This approach is more sophisticated than a simple classification problem and can help improve the accuracy of US examinations, even in challenging cases in which precise localization is difficult. We contribute as follows:We propose a scanning location-guided US system for accurate liver diagnosis that emphasizes the relationship between probe posture and the liver to improve accuracy and reduce diagnostic variability. Our method is the first to incorporate US-based scan location recognition for liver diagnosis by emphasizing the probe-posture and liver relationship.We develop a hierarchical deep learning architecture for liver scan location classification in US images by leveraging organ (liver, kidney, gallbladder) and vessel segmentation information and introducing a probabilistic representation method to enhance prediction robustness.We address the challenges of identifying key liver regions and handling ambiguous data, and we improve diagnostic accuracy in difficult cases by accounting for slight movement during US examinations.

## 2. Overview of the Proposed Approach

In this section, we provide an overview of the proposed approach, which aims to address the limitations of previous methods and enable the high-accuracy classification of liver US images into 11 standardized reference scan locations. Note that the 11 standardized scan locations are directly linked to reference probe postures that provide US images of liver segments important for diagnosis. Due to large variations and ambiguities in liver US images, including inter-subject variability, the classification of standardized scan locations in US images has yet to be achieved with sufficient accuracy.

First, we adopted the approach proposed by a previous work [7] in which a well-established VGG network [16] was used to classify LUSP images into eight classes. Based on random initialization-based training and transfer learning-based fine-tuning of the VGG network, those authors reported classification accuracy of about 92%. We implemented their approach to classify 11 standardized reference scan locations, but we replaced their VGG network with a ResNet-50 network [17], which is more powerful than a VGG network, as shown in Figure 1.

However, we observed that the performance of the ResNet-50-based classifier for 11 standardized scan locations, an average F1-score of 0.74, was not as good as the performance reported for the eight classes of LUSP images, possibly because the US images obtained from the 11 standardized locations are subject to high variability and complexity due to the use of various probe poses and individual differences. Therefore, we decided to look for a method to extract more powerful features that could deal with the complexity involved in classifying 11 standardized locations. Note that the LUSP classes are defined by specific feature sets present in the US images rather than by reference scan locations.

To improve the performance of our system, we chose to use a transformer-based segmentation network, MaskFormer [11], to extract more powerful feature sets from mask and pixel embedding and thereby improve classification, as shown in Figure 2.

In Figure 2, the MaskFormer segments organs (liver, kidney, and gallbladder) and vessels to provide a mask class and embedding, as well as segmentation output. We concatenated the predicted classes, mask embedding, and geometric features from the segmented organs and vessels with the image features from the MaskFormer encoder and used all of them as the input to the classifier. Based on the MaskFormer-based scan location classification network, we were able to obtain an F1-score of 0.87 when classifying the 11 scan locations, which represents an impressive performance improvement over fine-tuning of the ResNet-50 network by transfer learning. Despite that improvement, we found that some of the 11 classes, such as the substernal transverse (ST), right subcostal scan portal vein (RSPV), and right subcostal scan hepatic vein (RSHV) classes, as well as the substernal longitudinal (SL) and extrahepatic bile duct longitudinal (EBL) classes, showed particularly high confusion among themselves. For instance, the average F1-scores for the first group (ST, RSPV, and RSHV) and the second group (SL and EBL) were 0.85 and 0.72, respectively, but the average F1-score for the rest of the classes was 0.91.

To address that imbalance in performance, we came up with our final proposal, a MaskFormer-based hierarchical classification of US images into 11 scan locations, as shown in Figure 3. In this approach, the 11 scan locations are hierarchically classified. First, eight classes are made by combining ST, RSPV, and RSHV into one class and SL and EBL into another, and the images are divided into those classes. Then, the two combined groups are separated into their member classes in later steps. As shown in Figure 3, the eight-class classification in the first hierarchy uses the MaskFormer-based classifier shown in Figure 2 but only for the segmentation of organs. To classify SL and EBL in the second hierarchical step, the MaskFormer-based classifier, shown in Figure 2, is used based only on the vessel segmentation. The classification of ST, RSPV, and RSHV requires a more complicated process due to high ambiguity among those classes in US images. To solve that problem, the ST, RSPV, and RSHV classifier is implemented as a two-step classification process: (1) The MaskFormer-based classifier, shown in Figure 2, is trained using the liver and vessels in US images that clearly represent the individual classes. (2) Those class features are then represented in terms of reference points in the feature space such that the feature points from ambiguous US images can be classified based on their distances from nearby reference points. This hierarchical approach effectively resolves the confusion among challenging classes and significantly improves the classification performance for those cases, resulting in an additional improvement in the F1-score from 0.87 to 0.93. Specifically, for the ST, RSPV, and RSHV classes and the SL and EBL classes, the improvements in the F1-score were from 0.85 to 0.93 and from 0.72 to 0.91, respectively.

## 3. Datasets

The experimental dataset for 11 standardized scanning locations estimation was composed of two primary datasets: Hospital Dataset, used exclusively for organ segmentation, and the 5 Subject Dataset. The Hospital Dataset is a subset of an ethically approved dataset from Samsung Medical Center (SMC), featuring a diverse range of patients in terms of age and gender. This dataset was acquired using five different machines, SEQUOIA, iU22, LOGIQ, HI VISION Ascendus, and EPIQ 5G. The 5 Subject Dataset was obtained by recruiting five subjects and obtaining their informed consent. Radiologists directly collected the data at the 11 scan locations: SL, ST, RSPV, right subcostal scan-right liver transverse (RSLT), liver dome (LD), RSHV, gallbladder longitudinal (GBL), EBL, right intercostal oblique scan-anterior (RIA), right intercostal oblique scan-posterior (RIP), and liver-kidney (LK), as shown in Figure 4. A specialist at SMC performed six different motions (fanning, rotating, shaking, sliding, sweeping, and compressing) on the subjects for 10–15 s at each scanning location [18], capturing the images on a LOGIQ machine to ensure a variety of appearances.

## 4. Deep Hierarchical Network (DHN) for Standardized US Liver Scan (LS) Classification: LS-DHN

### 4.1. Feature Extraction from Segmentation Network

In this section, we introduce the feature extraction process from the segmentation networks, which plays a crucial role in our proposed liver scan location classification method. Figure 5 shows a visual representation of the MaskFormer-based feature extraction module. Note that the organ (liver, kidney, and gallbladder) and vessel segmentation models are trained independently.

#### 4.1.1. Global Long-Term Feature Extraction

In this section, we describe the process of extracting the global long-term features highlighted by the blue boxes in Figure 5 from both organs and the vessel segmentation networks, which are independently trained to classify liver scan locations in US images. These features are derived from the outputs of the ResNet-based feature pyramid network (FPN) [19] and the transformer decoder included in each segmentation model.

Image Feature Extraction: We apply global average pooling (GAP) to the output of the first layer of the FPN decoder for the semantic segmentation network to extract image features. Let F∈ℝCF×HF×WF be the output feature map, where CF is the number of channels, and HF and WF are the height and width of the input image divided by a stride of 32. The GAP operation is given by:(1)Gi=1HF×WF∑h=1HF∑w=1WFFi,h,w
where Gi denotes the GAP feature for channel i.

Transformer Decoder Feature Extraction: The pooling layer of a CNN has the disadvantage of losing important information and failing to encode the relative spatial relationships between feature maps. Using transformer decoder information, we expect to be able to effectively train for long-term dependencies. Therefore, the transformer decoder is used to produce N embeddings (Q) under the influence of the transformed image features. The Q embeddings independently generate N class predictions with N corresponding mask embeddings. We set N equal to the number of queries and equal to the number of classes to ensure a one-to-one correspondence between activations and classes during training. Let Q∈ℝCQ×N be the per-segment embeddings of dimension CQ that encode global information about each segment that the model predicts.

Geometric Feature Vector Generation: We create a binary mask using the output from the semantic segmentation, which contains the segmentation classes at each pixel, excluding the background class. Let F∈ℝN×H×W be the output of the semantic segmentation, where N represents the number of classes, including the background. We obtain a binary mask of size H×W by taking the argmax along the class dimension for each channel. We use that binary mask to generate vectors that represent each segmentation class, excluding the background class. Let V∈ℝ(N−1)×10 be the matrix of geometric features, with each row representing a single class. The components of each row are formulated as follows:(2)Tag:Ti={1,  if class i is detected0,  otherwise
(3)Location: (Li,x,Li,y)=(∑p∈Cixp|Ci|,∑p∈Ciyp|Ci|)
where Ci is the set of segmented pixels for class i, and (xp,yp) are the coordinates of pixel p.
(4)Size:Si=|Ci|H×W
(5)Morphology:Mi=[λ1,λ2,v1,v2]
where H and W are the height and width of the input image, respectively. λ1 and λ2 are the eigenvalues, and v1 and v2 are the eigenvectors of the detected binary mask for class i. Both the eigenvalues and eigenvectors are flattened into one-dimensional representations. For undetected classes, the corresponding vectors are filled with zeros. Thus, the geometric feature vector for class i is given by Vi=[Ti,Li,x,Li,y,Si,Mi].

Finally, we vectorize and concatenate all the values obtained in the above three components to use them as MLP inputs for the scan location classification network.

#### 4.1.2. Vessel-Related Feature Extraction

This section describes the process of extracting the vessel features from the vessel segmentation highlighted in the green box in Figure 5. The transformer decoder generates N per-segment embeddings (Q) by attending to the image features. Here N is 2, indicating the background and the vessel. These Q embeddings are used as input to an MLP, which independently generates N class predictions and N mask predictions. To obtain the mask predictions, we first apply the dot product between the per-pixel embeddings and the mask embeddings. Let P∈ℝN×H×W represent the mask predictions after this dot product operation, where H and W are the height and width of the input image, respectively. To use the mask predictions as an image, we need to eliminate negative values because they are included in the model’s output. When we apply the sigmoid function to the mask predictions, values below 0.5 are closer to 0, effectively suppressing lower-confidence predictions. To retain these lower-confidence values and filter out the negative ones, we apply the ReLU activation function to the mask predictions instead of the sigmoid function. Let P^∈ℝN×H×W represent the mask predictions after ReLU activation. The elements of P^ can then be formulated as:(6)P^n,h,w=max(Pn,h,w,0)
where Pn,h,w is the mask prediction before applying any activation function. Finally, we create a three-channel image I∈ℝH×W×3=concatenate (V,P1,P2), where V is the segmented vessel area of the vessel segmentation input image, and the rest of the area is filled with zeros; P1 is the mask prediction for the background; and P2 is the mask prediction for the vessel. This image provides a clear visual representation for further analysis and serves as the input for the ResNet-50 network.

### 4.2. Architecture of the LS-DHN

In this section, we present the architecture for hierarchically classifying liver US images into 11 scan locations. Figure 6 provides an overview of the hierarchical architecture, and the blue and green boxes correspond to the respective feature extraction methods shown in Figure 5, emphasizing that the same feature extraction techniques are used in the hierarchical architecture.

#### 4.2.1. Handling Ambiguous Data with Probabilistic Representation

A significant contribution of our proposed method is the effective management of ambiguous data, and novel liver US scan data from new subjects using probabilistic representation. In the hierarchical architecture, we first group similar regions and classes together to reduce the complexity of the problem and enable more focused and specific scan location classification. This helps to alleviate some of the ambiguity present in the data by narrowing down the potential class options for each liver scan location, as well as addressing the variability introduced by novel data from new subjects. After that initial classification, the correctly predicted data are referred to as clear data.

For the ST, RSPV, and RSHV classes, which represent the right lobe of the liver, the primary organ visible is the liver due to their similar locations. Consequently, the shape of the vessels becomes crucial for classification. However, as shown in Figure 7, ambiguous data pose challenges in distinguishing these classes.

We thus refine the classification process using the probabilistic representation of ambiguous data. We train an encoder with clear data using a triplet loss function, which helps to create a more discriminative representation of the liver and vessel segmentation features. Next, we create a 2D t-SNE embedding of clear data points and place a uniform 10 × 10 grid over its distribution to serve as a guide for identifying ambiguous data points. We create a set of reference points for each grid cell by uniformly sampling a clear data point from that cell. Finally, we determine the class labels for ambiguous data points by using the K-nearest neighbor (KNN) algorithm with a k value of 3 to find their distances from the reference points in the 64-dimensional embedding space, as shown in Figure 8.

#### 4.2.2. First Hierarchy: Classification of Eight Liver Regions

In our proposed method, we use a hierarchical approach to classify the 11 scan locations. First, we group ST, RSPV, and RSHV, which represent similar regions in the right lobe of the liver, into one class, and group the SL and EBL classes, which also have similar features, into another class, producing eight classes. To classify these liver regions, we use only features extracted from the organ (liver, kidney, or gallbladder) semantic segmentation model, as described in Section 4.1.1. Organ information alone is sufficient to identify the eight scanning locations.

#### 4.2.3. Second Hierarchy: Hierarchical Classification of the ST, RSPV, and RSHV Classes by Handling Ambiguous Data

We refine the classification of the ST, RSPV, and RSHV classes using features from the liver in the organ segmentation and vessel features from the vessel segmentation, as described in Section 4.1.1. The data points predicted correctly in that initial classification (Sub-step 1 in Figure 6) are considered to be clear data.

To create a more discriminative representation of the ambiguous data points, we train an encoder with clear data using a triplet loss function to ensure that an anchor image (a) is closer to a positive image (p) of the same class than to a negative image (n) of a different class by a preset margin (α):(7)$$L\left(a,p,n\right)=\max\{\left|a_{i}-p_{i}{|}^{2}-\right|a_{i}-n_{i}{|}^{2}+\alpha ,\;0\}$$

We train the encoder using online sampling and setting α to 1. After obtaining the 64-dimensional feature vectors of the clear data points, we generate 2D t-SNE embedding of those points. In the t-SNE space, we place a uniform 10 × 10 grid over the distribution and create a set of reference points for each grid cell by uniformly sampling one clear data point from each cell. This set of reference points serves as a basis for identifying ambiguous data points in the embedding space. We classify the ambiguous data points using KNN (k = 3) based on the 64-dimensional feature vectors. By handling ambiguous data using a probabilistic representation, we improve the robustness of the hierarchical classification approach for these classes.

#### 4.2.4. Second Hierarchy: Hierarchical Classification of the SL and EBL Classes

We next focus on the hierarchical classification of the SL and EBL vessel classes. Because the liver and vessel locations for these classes are similar, we rely on vessel segmentation features. We enhance the feature extraction process by using mask predictions from the vessel segmentation network, as described in Section 4.1.2. In Figure 9, the red channel represents the segmented vessel area in the original images from the vessel segmentation, highlighting the detected vessels. The blue channel represents the mask prediction for the vessels, highlighting the potential for the mask prediction to compensate for undetected vessels. The green channel highlights background areas without vessels. The extracted feature is input into the ResNet-50 network loaded with the pre-trained ImageNet weights, and all the layers are fine-tuned to distinguish the SL and EBL classes.

## 5. Results

In the experiments, we used a 5 Subject Dataset with a total of 23,611 images and an additional Hospital Dataset of 5659 images from 244 patients. For organ semantic segmentation, we used both the Hospital Dataset and the 5 Subject Dataset, whereas, for vessel segmentation and the scan location classification experiments, only the 5 Subject Dataset was used. We divided the dataset into three training subjects and two validation subjects for all experiments, performing 10-fold cross-validation to ensure inter-individual generalization. The optimizer used for the US scan location classification was AdamW, with the learning rate set to 0.0001. Additionally, the training was set to stop if the validation performance did not improve over 10 epochs, and it used an MLP layer consisting of two fully connected layers with the cross-entropy loss for the classification. We evaluated the performance of the liver US scan location classification using three key metrics: recall, precision, and F1-score.

### 5.1. Performance of Organ and Vessel Semantic Segmentation

We separately trained the organ (liver, kidney, and gallbladder) segmentation and vessel segmentation models using the MaskFormer network. For each model, we trained for 80,000 iterations. In the MaskFormer networks, we used a ResNet-50 backbone pre-trained on the ImageNet dataset and fine-tuned it during training, and the rest of the network was trained from scratch. The dataset used for organ segmentation includes the Hospital Dataset and the 5 Subject. For vessel segmentation, we used only data from the 5 Subject Dataset acquired in a sequence with scanning motion, as shown in Table 1.

### 5.2. Performance of MaskFormer-Based Liver US Scan Location Classification

In this section, we present a comparative analysis of liver US scan location classification performance from three distinct systems: ImageNet fine-tuned ResNet-50, Non-Hierarchical, and LS-DHN. Our primary focus lies in evaluating the efficacy of the proposed hierarchical architecture (LS-DHN), which has demonstrated superior performance. For a comprehensive comparison, we also provide an ablation study of the non-hierarchical architecture.

#### 5.2.1. Performance of the Hierarchical LS-DHN

**First Hierarchy—Classification Results for the Eight Liver Regions:** We grouped classes with similar locations (ST, RSPV, and RSHV) into a single class and combined classes with similar features (SL and EBL) into another class. We then used the organ segmentation information as input to an MLP for classification. The classification results for the eight liver regions are shown in Table 2. We achieved high precision, recall, and F1-scores for all classes.

**Second Hierarchy—Hierarchical Classification Results for the ST, RSPV, and RSHV Classes By Handling Ambiguous Data:** Because the ST, RSPV, and RSHV classes are detected in the liver, we use only the liver features from the organ segmentation, along with the vessels, which are significant in distinguishing the classes. Because these three classes correspond to similar locations in the right lobe of the liver, the locations become ambiguous in some US images due to motion during diagnosis. To address that issue, we experimented to improve the handling of ambiguous data. Processing the ambiguous data helped us to classify the scan locations more specifically (Table 3). The MLP layer used to create the embedding consists of three layers, with 64 final dimensions, batch normalization, and an activation function that uses Leaky ReLU.

**Second Hierarchy—Hierarchical Classification Results for the SL and EBL Classes:** Our method, described in Section 4.2.4, uses the image information of mask prediction to compensate for the low probability of vessel detection in the SL and EBL classes. The input images were resized to 224 × 224 × 3, and pixel values were normalized to a range between 0 and 1. This approach produced a significant improvement in the F1-score of the model’s predictions for the SL and EBL classes, which are often confused due to the presence of vessels and similar liver information (Table 4).

#### 5.2.2. Ablation Study: Non-Hierarchical Liver US Scan Location Classification

In this section, we present an ablation study conducted to evaluate the performance of the non-hierarchical approach to classifying liver US scan locations. This analysis allows us to compare the performance of our hierarchical architecture (LS-DHN) with a non-hierarchical architecture. The latter performs better than the ImageNet fine-tuned ResNet-50, but it is not as effective as the hierarchical approach. The non-hierarchical architecture classifies the 11 scan locations with a single model by simultaneously segmenting all the classes detected in the organs (liver, kidney, and gallbladder) and vessels. The feature extraction process for the segmentation model follows the methodology described in Section 4.1.1. After extraction, the features derived from the segmentation model are concatenated and classified by applying two fully connected layers.

As shown in Table 4, the non-hierarchical approach has an average F1-score of 0.87, which consistently outperforms the ImageNet fine-tuned ResNet-50, with its average F1-score of 0.74. However, our proposed LS-DHN achieves even higher performance with an average F1-score of 0.93, demonstrating the effectiveness of our hierarchical architecture in liver US scan location classification.

### 5.3. Discussion

In the experiment, we found that the photometric and geometric features extracted from segmented liver, kidney, and gall bladder images provide a high accuracy classification of several scan locations such as RSLT, LD, GBL, RIA, RIP, and LK, while they are largely ineffective for distinguishing the rest of the scan locations. Specifically, the rest of the scan locations can be grouped in such a way that those scan locations that belong to a group are heavily misclassified among themselves based only on the organ-based features, for example, ST, RSPV, and RSHV group, as well as the SL and EBL group. Subsequently, we found that those groups of scan locations that are unable to distinguish by the organ-based features only can be discriminated based on the features associated with the major blood vessels in the liver.

Indeed, by adding the photometric and geometric features extracted separately from the segmented major liver blood vessels, we could dramatically improve the accuracy in the classification of all 11 scan locations, as shown in Table 4. However, despite the dramatic improvement in accuracy, we found that some scan locations, such as EBL and SL, still need further improvement in accuracy and that those scan locations that show high accuracy with the organ-based features only are somewhat degraded in their accuracies with the addition of the features from segmented blood vessel images, as shown in Table 4.

To address this issue, we proposed to use a hierarchical approach in which the organ-based features are used in the first layer to classify out those scan locations and groups of scan locations, i.e., RSLT, LD, GBL, RIA, RIP, LK, (ST, RSPV, and RSHV) and (SL and EBL), that result in high accuracy in classification, while the two groups of scan locations, (ST, RSPV, and RSHV) and (SL and EBL), are classified separately in the second layer based on the approaches optimal for their unique needs (Refer to Figure 6).

In particular, we noticed that, for the (ST, RSPV, and RSHV) group, the photometric and geometric features associated with the liver and blood vessels could play an important role in classification whereas, for the (SL and EBL) group, only photometric features associated with blood vessels are significant. In addition, we observed that the three classes in the (ST, RSPV, and RSHV) group show a large portion of their images that are hard to distinguish, whereas the two classes in the (SL and EBL) group are relatively better in discriminating their images.

Based on these observations, we custom-designed the two classifiers optimal, respectively, for the (ST, RSPV, and RSHV) group and the (SL and EBL) group. In classifying the (ST, RSPV, and RSHV) group based on the photometric and geometric features associated with the liver and blood vessels, using a conventional deep learning architecture such as ResNet-50 was found not so effective for classification, as indicated in Table 4, due to inter-class similarities among a large portion of class images. To solve this problem, we devised a novel approach in which a number of typical image features are selected from individual classes as the representative features of the class such that the classification of an input image is done based on the voting by k-nearest representative features from the input image.

This approach is proven to be very effective in enhancing classification accuracy, as shown in Table 3. On the other hand, in classifying the (SL and EBL) group based on the photometric features associated with the blood vessels, we found that ResNet-50 works well for classification. However, to further improve classification accuracy, we concatenated the intensity of segmented blood vessel images with the probability of the predicted blood vessel masks as the input to ResNet-50 (Refer to Figure 9 and Table 4). As a result, we could obtain high classification accuracy for all 11 scan locations.

Note that the diverse and non-sequential nature of the real Hospital Dataset has a positive impact on the overall performance since the dataset provides different morphologies of the liver, kidney, and gallbladder to serve as the basis for the subsequent hierarchical classification.

## 6. Conclusions and Future Work

In this paper, we have presented a novel way to improve the F1-score of liver disease diagnosis using US images: the LS-DHN hierarchical classification architecture. Our proposed method can directly guide examiners to 11 scan locations for key liver regions during US examinations, thereby overcoming the challenges posed by patient variability and enabling more consistent diagnosis among patients. The step-by-step approach used in the hierarchical architecture enables more accurate and refined classification results, especially for challenging classes such as ST, RSPV, and RSHV, and SL and EBL.

We have also incorporated a probabilistic representation approach to handle ambiguous data, which enhances the robustness of our model’s predictions. This approach effectively addresses the challenges that arise from inter-patient variability, as well as imaging ambiguity and complexity caused by probe movement during diagnosis. Consequently, it facilitates more consistent and reliable diagnoses across diverse clinical scenarios, irrespective of individual patient differences or complications introduced by probe movement.

Our experimental results show that our proposed hierarchical architecture (LS-DHN) outperforms the non-hierarchical architecture and ImageNet Fine-Tuned ResNet-50, achieving higher scores for the overall classes and improving the performance of liver US scan location classification. The efficient processing times at each stage and the real-time capabilities of our system demonstrate the method’s potential for seamless integration into medical workflows. Our proposed method demonstrates real-time performance by efficiently processing liver US images at each stage of the pipeline. The entire pipeline takes approximately 154 ms per image for processing, which translates to approximately 6.49 frames per second on an RTX 3070 Ti GPU with a batch size of 1. This real-time performance ensures that our method can be effectively used for liver US in clinical settings, providing examiners with immediate feedback to facilitate accurate diagnosis and decision-making.

In future work, we plan to explore the potential of our proposed method in clinical practice by performing more comprehensive experiments on larger datasets and evaluating its ability to account for patient variability and improve the overall quality of liver disease diagnosis using US images.

## Figures and Tables

**Figure 1 sensors-23-04850-f001:**
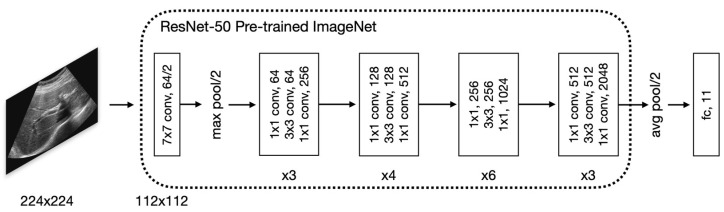
The model is fine-tuned on a ResNet-50 network and uses pre-trained ImageNet weights to classify images into 11 standardized reference scan locations.

**Figure 2 sensors-23-04850-f002:**
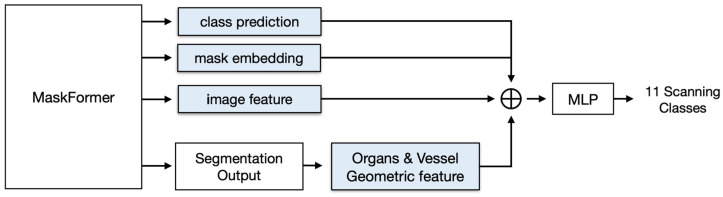
A schematic diagram of the MaskFormer-based scan location classification network.

**Figure 3 sensors-23-04850-f003:**
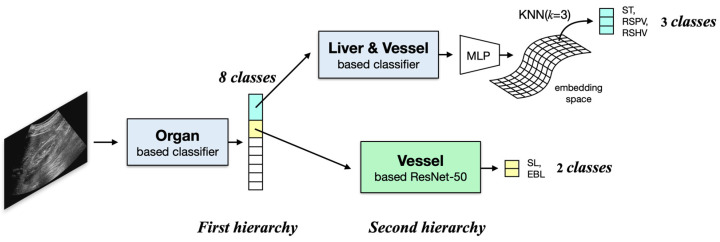
The hierarchical network architecture shows the process of grouping ambiguous classes and extracting features using separate segmentation of organs and vessels to improve classification performance. The blue boxes represent one feature extraction method, and the green box represents another feature extraction method.

**Figure 4 sensors-23-04850-f004:**
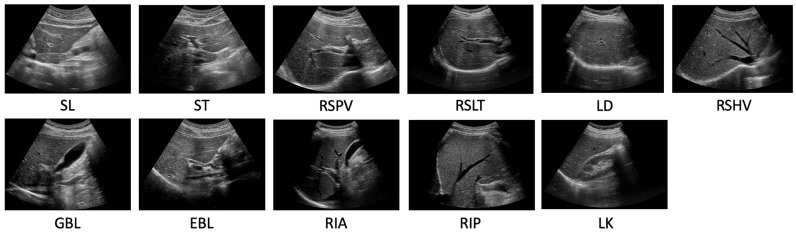
Representative images of the 11 standardized scan locations used for classification. These images were captured using a LOGIQ machine and six distinct examination motions at each scanning location to ensure a diverse range of appearances.

**Figure 5 sensors-23-04850-f005:**
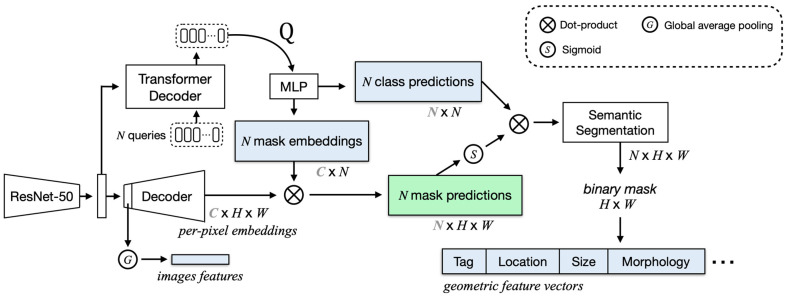
Segmentation-based (MaskFormer) feature extraction module: The blue and green boxes represent the two feature extraction methods used to classify liver scan locations.

**Figure 6 sensors-23-04850-f006:**
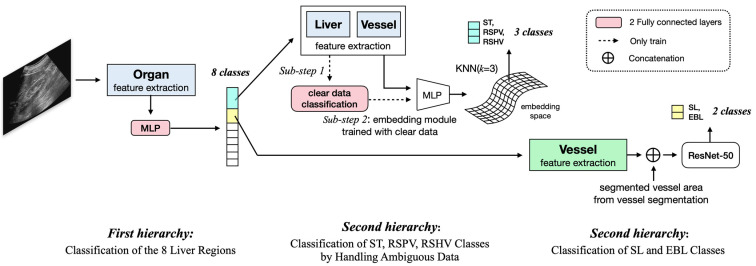
Overview of the hierarchical architecture for liver US scan location classification. The blue and green boxes within the figure correspond to the feature extraction methods in Figure 5, indicating the use of the same features.

**Figure 7 sensors-23-04850-f007:**
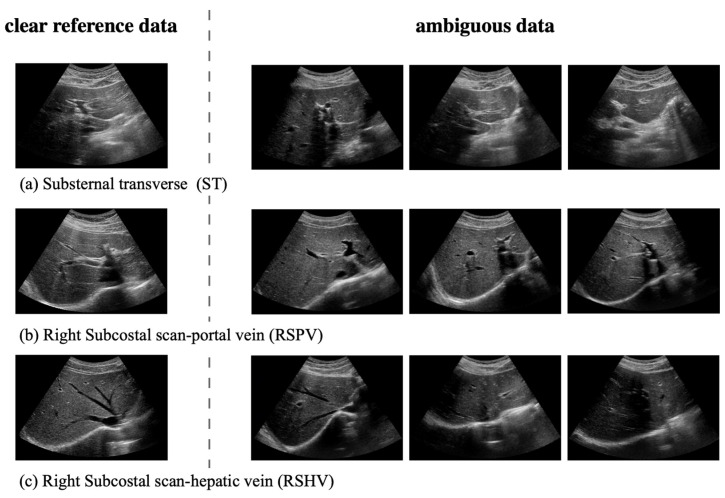
Examples of ambiguous liver US images. These images show that the characteristics and presence of vessels in the ambiguous data keep changing, making it difficult to classify them using the non-hierarchical architecture alone.

**Figure 8 sensors-23-04850-f008:**
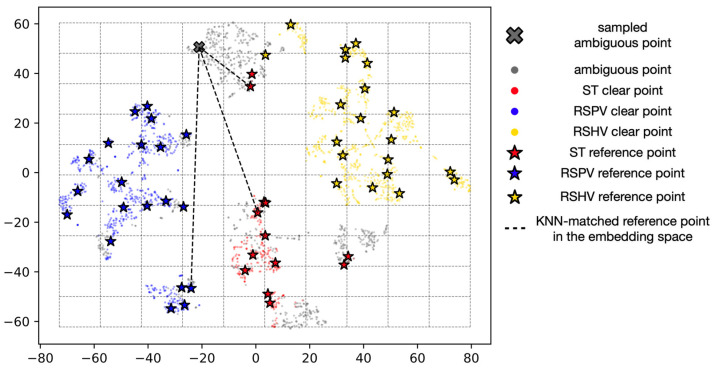
Visualization of ambiguous data handling using probabilistic representation. The t-SNE 2D plot shows clear data forming a grid. In the embedding space (64-d) for each ambiguous data point, the KNN algorithm selects the 3 nearest reference points from among all the reference points generated in each grid cell.

**Figure 9 sensors-23-04850-f009:**
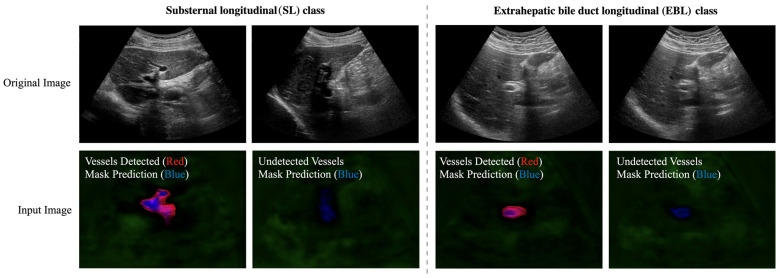
Visualization of the original and input images for distinguishing the SL and EBL classes. The red and blue channels represent segmented vessels and mask predictions, respectively. The blue channel emphasizes areas in which mask predictions compensate for undetected vessels.

**Table 1 sensors-23-04850-t001:** Performance of Liver, Kidney, Gallbladder, and Vessel Semantic Segmentation.

Model	Class	IoU	Pixel Acc
Organ	Liver	85.57	92.44
	Kidney	82.79	90.79
	Gallbladder	70.31	81.38
Vessel		50.80	63.40

**Table 2 sensors-23-04850-t002:** Classification Results for the 8 Liver Regions.

Method	w/Organ (Liver & Kidney & Gallbladder) Info		
Class	Precision	Recall	F1-Score
ST, RSPV, RSHV	0.97	0.99	0.98
SL, EBL	0.95	0.94	0.95
RSLT	0.94	0.86	0.90
LD	0.89	0.96	0.93
GBL	0.92	0.97	0.95
RIA	0.91	0.89	0.90
RIP	0.92	0.88	0.90
LK	1.00	0.99	0.99

**Table 3 sensors-23-04850-t003:** Classification results for the ST, RSPV, and RSHV classes using the hierarchical approach, highlighting the performance improvement achieved by handling ambiguous data in Sub-step 2 after the initial classification in Sub-step 1.

Method	(Before) Sub-Step 1: w/Liver & Vessel Info (after) Sub-Step 2: Probabilistic Representation w/Liver & Vessel Info		
Class	Precision	Recall	F1-score
ST	0.92 → 0.98	0.79 → 0.84	0.85 → 0.90
RSPV	0.88 → 0.95	0.90 → 0.98	0.89 → 0.96
RSHV	0.85 → 0.88	0.97 → 1.00	0.91 → 0.93

**Table 4 sensors-23-04850-t004:** Comparison of ImageNet Fine-Tuned ResNet-50, Non-hierarchical Architecture, and LS-DHN Performance.

Method	ImageNet Fine-Tuned ResNet-50			Non-Hierarchical			LS-DHN		
Class	Precision	Recall	F1-score	Precision	Recall	F1-score	Precision	Recall	F1-score
SL	0.56	0.81	0.66	0.74	0.88	0.81	0.88	0.98	**0.93**
ST	0.62	0.65	0.64	0.85	0.78	0.82	0.98	0.84	**0.90**
RSPV	0.73	0.61	0.66	0.91	0.85	0.88	0.95	0.98	**0.96**
RSLT	0.85	0.95	**0.90**	0.91	0.88	**0.90**	0.94	0.86	**0.90**
LD	0.88	0.84	0.86	0.91	0.97	**0.94**	0.89	0.96	0.93
RSHV	0.72	0.78	0.75	0.79	0.94	0.86	0.88	1.00	**0.93**
GBL	0.81	0.64	0.72	0.90	0.95	0.93	0.92	0.97	**0.95**
EBL	0.62	0.33	0.43	0.86	0.49	0.63	0.96	0.81	**0.88**
RIA	0.80	0.89	0.84	0.84	0.97	**0.90**	0.91	0.89	**0.90**
RIP	0.92	0.75	0.82	0.96	0.83	0.89	0.92	0.88	**0.99**
LK	0.85	0.89	0.87	1.00	0.99	**1.00**	1.00	0.99	0.99

## Data Availability

Not applicable.

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
