# Peer review of "Real-Time Deep Recognition of Standardized Liver Ultrasound Scan Locations"

_sensors, 2023, doi:10.3390/s23104850_

Round 1

Reviewer 1 Report

the paper is well-written, methods are very well-described, but the discussion lacks literature comparison or it is too fragmented. I advise you to split the results and the discussion to make reading easier. Furthermore, no  Gold Standard or comparison with it are mentioned. 

Reviewer 2 Report

Well organized research article with quality contribution from researchers. 

Few questions should be answered by the researchers here:

(1) How geometric features and Images features were selected? Elaborate and explain.

(2) Some related references should be cited in the text:

Das, T. K., Chowdhary, C. L., & Gao, X. Z. (2020). Chest X-ray investigation: A convolutional neural network approach. In Journal of Biomimetics, Biomaterials and Biomedical Engineering (Vol. 45, pp. 57-70). Trans Tech Publications Ltd.

(3) Diagrams and Tables are impressive.  

Reviewer 3 Report

The proposed methods are clearly introduced and verified by dataset. From the point of my view, this paper reaches the level of publishment in Sensors. There is one minor issue need to be solved.

The title of this paper is “Real-time Deep Recognition of Standardized Liver Ultrasound Scan Locations”. However, there seems no discussion about the “real-time” property. Please explain this concern.
